# Preparation and Characterization of Methyl Jasmonate Microcapsules and Their Preserving Effects on Postharvest Potato Tuber

**DOI:** 10.3390/molecules27154728

**Published:** 2022-07-24

**Authors:** Xiaozhen Han, Shuai Shao, Xiaocui Han, Yurong Zhang

**Affiliations:** Grain Storage and Security Engineering Research Center of Education Ministry, College of Food Science and Engineering, Henan University of Technology, Zhengzhou 450001, China; ss3211@163.com (S.S.); hanxiaocui0924@163.com (X.H.)

**Keywords:** potato, sprouting, quality, methyl jasmonate, microcapsule, TOPSIS

## Abstract

Potato tubers tend to sprout during long-term storage, resulting in quality deterioration and shortened shelf life. Restrictions on the use of chlorpropham, the major potato sprout suppressant, have led to a need to seek alternative methods. In this study, the effects of methyl jasmonate (MeJA) solutions and MeJA microcapsules on sprouting and other key quality attributes of the potato tuber were investigated. The results showed that the MeJA solution was most effective at 300 μmol L^−1^ according to TOPSIS analysis. To prepare MeJA microcapsules, the optimal formulation is with 0.04% emulsifier, 2.5% sodium alginate, 0.5% chitosan and 3% CaCl_2_. Compared to 300 μmol L^−1^ MeJA solution, MeJA microcapsules consumed a lower dose of MeJA but demonstrated a better retaining effect on the overall quality attributes of potato tubers. MeJA microcapsules are promising agents for the preservation of postharvest potato tubers.

## 1. Introduction

The potato is consumed worldwide and serves as an excellent source of nutrients such as carbohydrates, vitamins, minerals, and antioxidants. Due to its seasonal production, potatoes need to be stored for periods of up to months to supply the market continuously. During the long-term storage of potatoes, sprouting is considered a significant factor determining its quality and marketability [1,2,3]. Potato sprouting can not only lead to a significant increase in the toxic compounds of α-Solanine, but also cause color and texture change, weight and nutrition loss, and elevation of the reducing sugar that undermines the processing quality of potato [4,5,6]. Chlorpropham (CIPC) was once the most widely used anti-sprouting agent in potato storage management. However, its application has been challenged due to toxicity to the environment and humans, and thus a novel method of sprout control needs to be sought [7,8].

Jasmonates are naturally occurring phytohormones. Methyl jasmonate (MeJA) is an ester form of jasmonic acid, and has been demonstrated to possess higher efficacy in mediating intercellular communications [9]. Early studies reported that exogenous MeJA could induce potato tuber formation and indicate the involvement of JA in promoting dormancy development [10,11]. Then, MeJA was shown to reduce postharvest sprouting and improve the storage quality of radishes [12]. MeJA was also found to inhibit seed germination in a series of species [13,14,15]. In recent years, more and more studies have demonstrated that MeJA could enhance the quality attributes of postharvest agricultural products such as color, texture, nutrition, etc. [16,17]. 

Microencapsulation is a technology of embedding tiny particles of solid, liquid, or even gas materials in a semi-permeable or sealed polymeric matrix [18,19]. It has been widely used in food processing, medicine, cosmetics, and many other fields [19,20,21,22,23]. The significant advantage of this technology lies in its ability to enable the active ingredients in the core to be released in a controlled manner [18,19]. Alginate/chitosan is one of the most popular food-grade and biodegradable polymeric systems for microencapsulation [24,25,26]. 

To date, plant hormones like abscisic acid, ethylene, cytokinin, gibberellin, and auxin have been shown to be involved in the regulation of potato sprouting [3,27]. As a new type of plant hormone, the role of MeJA has been poorly investigated. In addition, as plant hormones are unstable and extremely sensitive to environmental changes such as temperature, humidity, and light [28], it is necessary to explore a method to enhance their stability and control their release. Therefore, the present study investigated whether MeJA could control sprouting and preserve the commercial qualities of stored potatoes, and established a safe and economical method using microencapsulation technology to maximize its effectiveness. 

## 2. Results and Discussion

### 2.1. Effect of MeJA Solution on the Quality Attributes of Stored Potato

We first investigated the effects of different concentrations of MeJA solution (10 μmol L^−1^, 50 μmol L^−1^, 100 μmol L^−1^, 200 μmol L^−1^, 300 μmol L^−1^, 500 μmol L^−1^, 1000 μmol L^−^^1^) on the sprouting index and other quality indexes (solanine content, greening index, hardness, weight loss, and dry matter) of potato tubers during storage.

The result showed MeJA solution could significantly delay potato sprouting and maintain the quality, but it seems these effects are dependent on the concentration of the MeJA solution (Figure 1). To comprehensively evaluate the preservation effects of MeJA solution at different concentrations on postharvest potatoes, the TOPSIS method was used. The 300 μmol L^−1^ MeJA achieved the highest TOPSIS score in terms of the overall effectiveness on potato preservation (Figure 1g). The TOPSIS score showed increasing trends with the increase in MeJA concentrations at ranges between 10 μmol L^−1^ to 300 μmol L^−1^, but started to decrease as MeJA concentration increased from 300 μmol L^−1^ to 1000 μmol L^−1^ (Figure 1g). 

The positive effect of MeJA on maintaining dormancy has been reported in previous studies (Wang 1998; Bazabakana et al., 1999; Blake et al., 2002; Rubio-Moraga et al., 2014). It was also shown this effect is more significant with the increase in MeJA concentration [12]. In correspondence with this study, we also noticed such an effect of MeJA on potato tubers, but we further found that countereffect might occur if MeJA concentration is too high. These antagonistic effects of MeJA were also reported in *Dioscorea alata* [29]. In fact, in addition to MeJA, R-carvone was also reported to have both inhibitory and catalyzing effects on the sprouting of potato tubers depending on the dosage [30]. 

Moreover, the preservation effect of MeJA solution was more prominent in the early stage than in the later stage of storage (Figure 1). The reason might be because MeJA volatilizes off at ambient temperature; thus, the effectiveness of MeJA decreased in the later stage of storage.

### 2.2. Formulations for the Preparation of MeJA Microcapsules

MeJA is a volatile ester and easily diffuses into the air, thus compromising or even losing its preserving effects. To overcome this disadvantage, we embedded emulsified MeJA into biopolymers of chitosan and alginate crosslinked with CaCl_2_.

The role of the emulsifier is to mix the hydrophobic MeJA evenly with water to facilitate the following encapsulation process with the alginate-chitosan. As reported in Rowe [31], the addition of an appropriate amount of emulsifier could significantly reduce the particle size of the emulsion. Our results showed the average particle size of the MeJA emulsion was 282.15 nm when there was no emulsifier (Figure 2a); with the addition of an emulsifier, the size of the MeJA emulsion decreased rapidly and then began to stabilize (Figure 2a). At a concentration of 0.04%, the average particle size was 75.33 nm, the smallest (Figure 2a). Since it has been found that a smaller particle size of emulsions is conducive to higher encapsulation efficiency and storage stability for the resulted microcapsules [32,33], 0.04% emulsifier was adopted. 

Sodium alginate is a linear polymer of b-D-mannuronic acid and a-L-guluronic acid from brown seaweed. Chitosan is a partially deacetylated polymer of N-acetylglucosamine obtained after alkaline deacetylation of chitin from crustacean shells. Both of them are food grade and have outstanding biocompatibility. However, compared with either polymer alone, their polyelectrolyte complex, formed by the electrostatic attraction between the cationic amino groups of chitosan and the carboxylic groups of the alginate, is more widely used with enhanced stability. As it has been reported previously that chitosan, alginate, and CaCl_2_ concentrations affect the microcapsule morphology and encapsulation efficiency [34,35], we designed a single factor and orthogonal experiment to optimize the ratio of chitosan, alginate, and CaCl_2_. The embedding rate of MeJA microcapsules significantly increased as the sodium alginate concentration increased from 1.0% to 2.5% (Figure 2b). However, the increase in the embedding rate became subtle when the sodium alginate concentration increased to 3% (Figure 2b). Besides, excessive sodium alginate would lead to the tailing phenomenon; thus, 2.0%, 2.5%, and 3.0% sodium alginate were chosen for the orthogonal experiment. CaCl_2_ is used because Ca^+^ can react with sodium alginate to promote the formation of a more stable eggshell-like structure and slow down the release of the core material MeJA. The embedding rate of MeJA increased with CaCl_2_ concentration and seemed to be maximized and stabilized when the CaCl_2_ concentration reached around 3.5–4.0% (Figure 2c). Thus 2.5%, 3%, and 3.5% were chosen as the three levels for CaCl_2_ in the orthogonal experiment. The embedding rate increased with the concentration of chitosan in the range between 0.1% and 0.5%, but changed little when the concentration was beyond 0.5% (Figure 2d). That might be because calcium alginate was not sufficiently sealed with a lower dosage of chitosan, but achieved saturation when chitosan was excessive. Thus 0.3%, 0.5%, and 0.7% were selected as the three levels of chitosan for the orthogonal test.

In the orthogonal experiment, the embedding rate of the MeJA microcapsule was taken as the evaluation index (Table 1). The results showed that the concentration of sodium alginate was the most influencing factor for the encapsulation of MeJA, and the optimal combination of the three factors was 2%/2.5% sodium alginate, 3.5% CaCl_2_, and 0.1% chitosan (Table 1).

### 2.3. Characterization of MeJA Microcapsules

At the magnification of 10.0 k, the prepared MeJA microcapsules are spherical with a diameter of around 2–3 μm, compact, and have relatively smooth surfaces without any cracking or tailing (Figure 3a), being the typical morphology for the polymer network of integrity that has been suggested in other studies [36,37,38]. These results indicate the freeze-drying process in this study was appropriate for producing good-quality MeJA microcapsules. The few protrusions or holes on the surface of the microcapsules may be caused by a small amount of MeJA wrapped by sodium alginate and chitosan being volatilized during the drying process. 

FT-IR is the technique used to obtain the infrared spectrum of absorption or emission of a solid, liquid, or gas. The infrared spectrum recorded by the detector generally ranges from 4000 to 400 cm^−1^, representing the sample’s unique molecular fingerprint. The development of MeJA microcapsules was further confirmed by FT-IR. The infrared spectra of MeJA, the mixture of alginate and chitosan, blank alginate-chitosan microcapsules, and MeJA microcapsules were examined. In (Figure 3b(A)), absorption peaks were observed at 2960 cm^−^^1^ and 1741 cm^−1^, which seem to be respectively attributed to the -CH (saturated alkanes) and C=O (ketones, aldehydes, and esters) in MeJA. Although the FTIR spectrum of the blank alginate–chitosan microcapsule in this study was slightly different from previous studies [39,40], it has the characteristic peaks around 1650 cm^−1^ and 3500 cm^−1^ (Figure 3b(C)), which are respectively assigned to the alginate carboxylic groups and the -OH and -NH2 in chitosan [36,41,42]. The subtle difference in the spectrum from other studies might be due to the varying concentrations of alginate and chitosan used. In contrast with the blank alginate–chitosan microcapsules, the characteristic absorption peaks of MeJA appeared in the infrared spectrum of MeJA microcapsules (Figure 3b(D)), indicating the successful embedment of MeJA in MeJA microcapsules. However, the absorption intensity at 2960 cm^−1^ became lower and the peak position around 1741 cm^−1^ was shifted to 1689 cm^−1^ (Figure 3b(D)), which might be due to molecular interaction between sodium alginate and MeJA, a phenomenon occurring during polymerization reported in many studies [39,43,44]. MeJA, blank alginate–chitosan microcapsules, and MeJA microcapsules exhibit broad absorption peaks around 3442 cm^−1^ (Figure 3b(A–D)), which could be assigned to the stretching vibration of O–H in MeJA or/and sodium alginate.

The compounds volatilized from MeJA microcapsules at day 0, 15, and 30 of storage were extracted by headspace solid-phase microextraction and analyzed by GC/MS. The characteristic absorption peaks of MeJA appeared after 8.11 min, and the peak area at day 0 was larger than day 15 and day 30 (Figure 3c), the reason for which might be due to a small amount of MeJA unembedded on the surface of the microcapsules being volatilized in the early stage. The peak intensity of MeJA at day 15 and day 30 was the same, and no other volatile substances were produced (Figure 3c(B,C)), indicating that MeJA microcapsules enable the specific and stable release of MeJA.

### 2.4. Release Capacity of MeJA Microcapsules

At 20 °C, the unencapsulated MeJA decreased by 35% in the first 25 h, and was close to zero after 120 h (Figure 4). In contrast, the amount of MeJA volatilized from MeJA microcapsules in the first 24 h was less than 5%, and the residual amount of MeJA was up to 80% after 120 h and 60.15% after 240 h (Figure 4). Besides, MeJA content in MeJA microcapsules decreased linearly over the entire 240 h examined (Figure 4). These results indicate the MeJA microcapsules possess a sustainable and stable release capacity of MeJA, and verify that the sodium alginate and chitosan system is a good system for controlled release of encapsulants as demonstrated by previous studies [45,46,47]. 

### 2.5. The Efficacy of MeJA Microcapsule on Quality Preservation of Stored Potatoes

The control samples began to sprout slowly right after being put into storage, and the sprouting rate increased rapidly from the 15th day, soaring to 98.2% at day 60 (Figure 5a). Treatments with 300 μmol L^−1^ MeJA and MeJA microcapsules both delayed sprouting for 15 d, but MeJA microcapsule-treated potatoes showed a significantly lower sprouting rate than 300 μmol L^−1^ MeJA treated samples as the storage time was prolonged (Figure 5a). This might be due to the rapid volatilization of MeJA for the application with 300 μmol L^−1^ MeJA solution at the later stage of storage, causing a weaker inhibition effect compared to MeJA microcapsules, which released MeJA steadily. Potatoes treated with CIPC started to sprout after 30 d of storage, with the sprouting rate being the lowest among all the samples, at only 18% after 75 d of storage (Figure 5a). 

The solanine contents of potato tubers treated with CIPC, 300 μmol L^−1^ MeJA, and MeJA microcapsules increased slower than in control samples (Figure 5b). The original solanine content of potatoes was 0.04 g kg^−1^, and reached 0.29 g kg^−^^1^ after 75 d for the control samples, 0.17 g kg^−1^ for the samples treated with 300 μmol L^−1^ MeJA, 0.13 g kg^−1^ for the samples treated with MeJA microcapsules, and 9 g kg^−1^ for samples treated by CIPC (Figure 5b). For solanine accumulation, it seemed CIPC had the strongest inhibition effect. In contrast, 300 μmol L^−1^ MeJA had a similar effect at the early stage of storage but a lower effect at the later stage than MeJA microcapsules.

The weight loss of control samples increased dramatically over the entire storage period, and reached up to 16.32% in 75 d (Figure 5c). In contrast, the weight losses of potatoes treated with 300 μmol L^−1^ MeJA, MeJA microcapsules, and CICP increased more slowly, and reached up to 9.63%, 9.02%, and 8.79% respectively after 75 d of storage (Figure 5c). 

The hardness of control potatoes decreased the most quickly from the original 80.5 N to 69.0 N after 75 d of storage (Figure 5d). The hardness of potato tubers treated with MeJA microcapsules was the same as that treated with 300 μmol L^−1^ MeJA from 0 to 45 d, but then decreased less at the later stage of storage (Figure 5d). After 75 d, the hardness of potato tubers treated with 300 μmol L^−1^ MeJA and MeJA microcapsules fell to 73.0 N and 75.8 N, respectively (Figure 5d). Interestingly, the hardness of potato tubers treated by CIPC decreased the second-most quickly and was 70.4 N after 75 d (Figure 5d). 

The a* value was measured to reflect the greening of potato tubers. The a* showed trends toward the negative Y-axis with the passage of storage time for all the samples, which indicates the occurrence of greening. Compared to the control, the greening of potato tubers treated with CIPC, 300 μmol L^−1^ MeJA, and MeJA microcapsules was slower (Figure 5e). The original a* value was 83.4; after 75 d of storage, it fell to 42.69 for the control, 64.76 for potato tubers treated with 300 μmol L^−1^ MeJA, 64.23 for potatoes treated with MeJA microcapsules, and 60.46 for potatoes treated with CIPC (Figure 5e). 

During the first 30 d of storage, the chlorophyll contents of all the potato samples changed little. After 30 d, the chlorophyll content of control samples started to increase rapidly from 0.02 mg kg^−1^ to 0.16 mg kg^−1^ at day 75 (Figure 5f). In contrast, the chlorophyll contents of potato tubers treated with 300 μmol L^−1^ MeJA, MeJA microcapsules and CIPC increased more slowly, reaching to 0.09 mg kg^−1^, 0.07 mg kg^−^^1^, 0.06 mg kg^−1^ respectively after 75 d (Figure 5f). 

The dry matter contents of all the samples showed decline trends during storage (Figure 5g); the most considerable decrease from 25.2% to 14.42% was observed in the control samples (Figure 5g). In contrast, the dry matter of potato tubers treated with 300 μmol L^−^^1^ MeJA, MeJA microcapsules and CIPC decreased by 8.39%, 6.97%, and 6.96% respectively over the examined storage time (Figure 5g). 

The reducing sugar contents of all the samples first exhibited increases from 0.35% and then dramatic decreases, peaking at the 30th day of storage (Figure 5h). The peaking reducing sugar contents were 0.38%, 0.37%, 0.37%, 0.39% for control, potato tubers treated with 300 μmol L^−1^ MeJA, MeJA microcapsules, and CIPC respectively (Figure 5h). After 75 d of storage, the reducing sugar contents decreased to 0.29%, 0.30%, 0.31%, 0.32% for the control and potato tubers treated with 300 μmol L^−1^ MeJA, MeJA microcapsules, and CIPC respectively (Figure 5h). 

The starch contents showed almost linear declines for all the samples (Figure 5i). The starch content of control samples decreased by 12.98% after 75 d of storage, while that of potato tubers treated with 300 μmol L^−1^ MeJA, MeJA microcapsules, and CIPC decreased by 11.98%, 10.48%, and 10.93% respectively (Figure 5i). 

Soluble protein contents decreased with the storage time for all the samples, and the decreases in potato tubers treated with CIPC, 300 μmol L^−1^ MeJA, and MeJA microcapsules were slightly slower than that in control samples (Figure 5j). Over 75 d of storage, the soluble protein contents of control samples, potato tubers treated with 300 μmol L^−1^ MeJA, MeJA microcapsules, and CIPC decreased by 1.02%, 0.92%, 0.78%, 0.89% respectively (Figure 5j). 

The contents of Vc decreased in a nearly linear manner with the storage time (Figure 5k). After 75 d of storage, the Vc contents of control samples, potato tubers treated with 300 μmol L^−1^ MeJA, MeJA microcapsules, and CIPC decreased by 0.15%, 0.13%, 0.12%, 0.07% respectively (Figure 5k). 

Recently, the effects of MeJA on the quality attributes of postharvest fruits have been widely studied [17]. Some studies demonstrated that exogenous application of MeJA could reduce fruit weight loss and softness, and increase the levels of soluble sugars and certain antioxidants such as vitamin C [48,49,50,51,52,53,54]. Similarly, MeJA-treated potato tubers also exhibited higher firmness, elevated contents of reducing sugar and vitamin C and lower weight loss in our study. Besides, it was observed in previous studies that potato tuber sprouting is often concurrent with the accumulation of toxic α-solanine, greening, firmness dropping, starch degradation, loss of fresh and dry weight, and agents that inhibit sprouting could also differentially slow down these processes [4,5,6], a phenomenon that was also observed in our study.

To comprehensively evaluate the results of various quality indexes, the TOPSIS method was used to analyze the effects of 300 μmol L^−1^ MeJA, MeJA microcapsules, and CIPC. From (Figure 5l), it can be seen that the MeJA microcapsules achieved the second-highest TOPSIS score for the efficacy on the preservation of the overall quality of stored potatoes, only being slightly lower than CIPC treatment. Although the TOPSIS score of 300 μmol L^−1^ MeJA treatment is dramatically higher than that of the control, it is significantly lower than MeJA microcapsules (Figure 5l). 

The dosage of MeJA by applying MeJA microcapsules to treat potato tubers is only 0.05 times the amount consumed by spraying 300 μmol L^−1^ MeJA solution. Even so, MeJA microcapsules have a slightly higher TOPSIS score than 300 μmol L^−^^1^ MeJA on the comprehensive evaluation of efficacy. Compared with direct spraying, the advantage of encapsulation lies in its ability for constant replenishment and thus avoiding fast deprivation or degradation so that the positive effects are extended, in accordance with similar cases reported in Ni, Acharya, Ren, Li, Li and Li [47], Mohammadi, et al. [55], and Taheri, et al. [56]. 

## 3. Materials and Methods

### 3.1. Materials

Fully mature potatoes (*Solanum tuberosum* cv. Atlantic) were freshly harvested from the field, and those with similar sizes between 50 and 60 mm in diameter and no mechanical damage were selected. 

### 3.2. Preparation of MeJA Microcapsules by the Two-Stage Piercing Method

#### 3.2.1. Procedures for MeJA Microcapsules Preparation

MeJA was dispersed into ultrapure water with the aid of the emulsifier under constant magnetic stirring. When the particle size of the formed emulsion approaches to nanometer size, 5 mL was added into 100 mL sodium alginate solution with continuous stir for 50 min. Then, the colloid was dropped into CaCl_2_ solution with the 10 mL syringe at the rate of one droplet per second with constant stirring for 30 min, followed by washing; the last step is to immerse in chitosan solution for 5 h to solidify until the double-layered MeJA microcapsules form.

#### 3.2.2. Encapsulation Efficiency of Microcapsules

MeJA was dissolved in 1% absolute ethanol. The solution was then scanned over the entire wavelength range by UV-Vis (TU-1810, Puxi General Instrument Co., Ltd., Beijing, China) spectrophotometer to determine λ_max_. The standard curve was plotted with the absorbance of MeJA at λ_max_ against its concentration.

The surfaces of the prepared microcapsules were washed with 100 mL distilled water three times to remove the residual calcium ion and MeJA. Then, 2 g microcapsules were ground and transferred into 10 mL absolute ethanol, followed by ultrasonic crushing for 30 min. After centrifugation at 4000× *g* for 5 min, the absorbance of the supernatant was determined and used to calculate the amount of MeJA in microcapsules according to the standard curve. Microcapsules without MeJA loading were used as control. The encapsulation efficiency of microcapsules was calculated with the equation below: $$w=\left(w_{1}-w_{2}\right)/w_{0}$$

*w*_0_—Initial addition amount of MeJA

*w*_1_—The amount of MeJA in MeJA microcapsules

*w*_2_—The amount of MeJA in the control microcapsules

#### 3.2.3. Single-Factor Optimization Experiment

Tween-80 in anhydrous ethanol (3:1, *w*/*w*) was used as the emulsifier. To optimize its concentration, the particle sizes of the resulted emulsions containing 0, 0.01%, 0.02%, 0.04%, 0.08%, and 0.10% emulsifiers were assessed by a particle size analyzer, and the concentration that leads to the minimum particle size is considered to be optimal. 

The concentration of sodium alginate solution was experimented with 2.5%, 3.0%, 3.5%, and 4.0%, while CaCl_2_ solution was set at 3% and the chitosan solution at 0.5%. The concentration of CaCl_2_ solution was experimented with 1%, 3%, 5%, and 7%, while sodium alginate solution was set at 3% and the chitosan solution at 0.5%. The concentration of chitosan solution was experimented with 0.1%, 0.3%, 0.5%, and 0.7%, while CaCl_2_ solution was set at 3% and the sodium alginate at 3%. The encapsulation efficiency was evaluated for these single-factor optimization experiments. 

#### 3.2.4. Orthogonal Optimization Experiment

Based on the results of single-factor optimization experiments, an orthogonal experiment with three factors and three levels was carried out. The three factors were the concentrations of sodium alginate, CaCl_2_, and chitosan solution. The three levels of sodium alginate solution were 1.5%, 2.0%, 2.5%, while that of CaCl_2_ were 2.5%, 3%, 3.5% and that of chitosan solution were 0.1%, 0.3%, 0.5%. The combination that leads to the highest encapsulation efficiency was considered the optimal formulation.

### 3.3. Characterization of MeJA Microcapsules

#### 3.3.1. Microscopic Observations

The morphology of MeJA microcapsules was observed by a scanning electron microscope (TM3000, Tianmei Scientific Instrument Co., Ltd., Shanghai, China).

#### 3.3.2. FTIR

The microcapsules were mixed with potassium bromide after grinding and pressed into tablets. The FTIR spectrum of microcapsules was determined by Fourier-transform infrared spectrometer (IR Prestige-21, Shimadzu, Kyoto, Japan). The step width was 2 cm^−1^, and the wavelength range was 4000–400 cm^−1^.

#### 3.3.3. GC-MS Analysis of MeJA Release

The volatilization of MeJA microcapsules was determined at 0, 15, and 30 d by headspace solid-phase microextraction/gas chromatography-mass spectrometry. 2.0 g MeJA microcapsules were placed in a 15 mL headspace sample bottle and sealed. Headspace extraction was conducted in a water bath at 60 °C for 60 min. Then the extraction head was pulled out from the sample bottle and inserted into the injection port of GC-MS (QP2010 Ultra, Shimadzu, Kyoto, Japan), being desorbed at 260 °C for 3 min for subsequent analysis.

GC-MS analysis conditions: DB-5 quartz capillary column (30 m × 0.32 mm × 0.5 μm) was used. The initial temperature was 50 °C for 4 min, then increased to 250 °C at the rate of 10 °C per min and kept for 15 min. The injection temperature was 280 °C, and the carrier gas was helium with a flow rate of 1.5 × 10^−3^ L per min. Mass spectrometry analysis conditions: electron impact (EI) source, 70 eV full scan, scanning range of 29~540 aum.

#### 3.3.4. Release Capacity of MeJA Microcapsules

Two grams of MeJA and two of MeJA microcapsules were put, and then kept in an incubator with a temperature of 20 °C and relative humidity of 80%. The residual MeJA in stock solution and microcapsules were measured every three hours. 

### 3.4. Application of MeJA Solution and Microcapsules

With sprayers, 0.03 g kg^−1^ CIPC and MeJA solutions at different concentrations were sprayed evenly onto the surfaces of the potatoes. The treated potatoes were first air dried, and then stored in a chamber with the temperature at 25 °C and relative humidity at 80%. Samples were taken every 15 d for the measurement of quality traits. 

Postharvest potatoes were stored in woven bags with small holes for air circulation, and MeJA microcapsules were placed inside with 0.1 g per 1 kg potatoes. The dosage of MeJA using this method is around 50.5 μg for treating 1 kg potatoes, only 0.05 times the amount consumed by spraying 300 μ mol L^−1^ MeJA solution.

### 3.5. Quality Parameters

#### 3.5.1. Sprouting Rate

According to Jia, Xu, Guan, Lin, Brennan, Yan and Zhao [6] ENREF_22, the sprouting rate was expressed as the ratio between sprouting eyes and the total number of eyes on the tuber.

#### 3.5.2. Solanine

The content of solanine was determined with the method described by Dong, et al. [57] with several modifications.

Extraction: 5.0 g of peeled potato pulp homogenized in a blender was first extracted with 50 mL of 70% methanol by ultrasonic method at 60 °C for 1 h, and then subjected to static extraction for 12 h. After filtration, the filtrate was concentrated into a paste in a rotary evaporator and then dissolved into 5 mL 5% sulfuric acid, followed by the adjustment of pH to 10.5 by aqua ammonia. After standing overnight in the refrigerator at 4 °C, solanine is entirely precipitated by centrifugation at 10,000× *g* for 20 min. The obtained residue was washed with 1% ammonia water, air dried, dissolved in 2 mL methanol containing 0.1% sulfuric acid, and then filtered through a 0.45 μm membrane filter for HPLC analysis (Agilent 1200, Agilent Technologies, Santa Clara, CA, USA).

The column in the HPLC system was Inertsil-ODS-3 (150 × 4.6 mm, 5 μm). The mobile phase consisted of acetonitrile and 0.02 mmol L^−1^ potassium monophosphate (25:75, *v*/*v*), and the flow rate was 1 mL per min. Absorbance at 208 nm was monitored by a variable wavelength scanning ultraviolet detector (G1314B, Agilent Technologies, Santa Clara, CA, USA). The solanine content was calculated by reference to a standard curve of pure solanine.

#### 3.5.3. Greening

The degree of greening can be reflected by the greenness of potato skin. In the CIELAB colorimetric system, a* ranges from green (−a*) to red (+a*) and can be used to evaluate greening. Five potatoes were measured at four different locations on each potato for each treatment. The potato epidermis with a thickness of 0.2 mm was sliced off, and then a* value was measured. Greening index in Figure 1 was measured starting from 20 June (Spring Potato), while Greening index in Figure 5 was measured starting from 5 November (autumn potato).

#### 3.5.4. Chlorophyll

The chlorophyll content in 2 mm thick potato skin was measured according to the method presented in Dong, Meng, Shi, Jiang and Wang [57] with some modifications. Two grams of sample was ground into homogenate with 10 mL 95% ethanol and 0.1 g calcium carbonate. Subsequently, the mixture was filtered and left standing for 3–5 min. The absorbance of the supernatant at 665 and 649 nm were measured with a UV-spectrophotometer and used to calculate the chlorophylls a and b contents in the extracts. Chlorophyll in Figure 5 was measured starting from November 5th (autumn potato).

#### 3.5.5. Hardness

After removing a thin slice of potato skin, hardness was recorded on two sides of each potato. A penetrometer (MH321, Puxi General Instrument Co., Ltd., Beijing, China) fitted with an 8 mm diameter flat probe was used to determine the hardness, which is indicated by the force required to overcome the penetration resistance by the plunger at a speed of 1 mm s^−1^ and expressed as N.

#### 3.5.6. Weight Loss

Weight loss was calculated according to the following formula: Weight loss (%) = (*W*_0_ − *W_n_*)/*W*_0_ × 100, where *W*_0_ was the initial weight of potato tubers, while *W_n_* was the weight of potato tubers during storage.

#### 3.5.7. Dry Matter Content

Two grams of thin-cut potato chips made from fresh potatoes were dried in the oven at 105 °C to a constant weight, recorded as the dry weight. Dry matter content was defined as the dry weight of samples expressed as a percentage of the fresh weight.

#### 3.5.8. Reducing Sugar

Reducing sugar was measured by the 3,5-dinitrosalicylic acid method [58] with some modifications. One gram of homogenized potato pulp was mixed with 5 mL distilled water in a centrifuge tube and then kept at 60 °C in the water bath for 20 min. After centrifugation at 2000× *g* for 10 min, 2 mL supernatant was transferred into a 25 mL tube and added with 1.6 mL 3,5-Dinitrosalicylic acid. The mixed solution was then heated at 100 °C for 5 min and cooled to ambient temperature. Then, the absorbance of the solution was measured at 540 nm with a spectrophotometer. The reducing sugar concentration is calculated based on a calibration curve of absorbances versus different glucose concentrations.

#### 3.5.9. Starch

Five grams of homogenized potato pulp was filtered with a folded filter paper in a funnel, and then washed with 150 mL ethanol (85%) to eliminate the soluble sugar [59]. The powder remaining on the filter paper was dried and completely transferred into a 250 mL beaker for starch determination. The starch content was measured according to Mccleary, et al. [60] with some modifications. Starch in the sample was first hydrolyzed to maltodextrins with α-amylase, and maltodextrins were further hydrolyzed with hydrochloric acid at 100 °C to glucose. Glucose was then determined using the Fehling’s solution method [61].

#### 3.5.10. Soluble Protein

The soluble protein was extracted by 5 mL distilled water from 2 g homogenized potato pulp. The soluble protein content was determined by absorbance at 595 nm according to the method of Bradford [62], using bovine serum albumin (BSA) as a standard.

#### 3.5.11. Vitamin C

The 2,6-dichloroindophenol titration method was used to detect the vitamin C content described in Ojukwu and Nwobi [63]. Five grams of potato pulp was homogenized in 2% metaphosphoric acid (*w*/*v*) and 2% oxalic acid (*w*/*v*). After centrifugation, 3 mL supernatant and 3 mL deionized water (blank) were titrated with the standard 2,6-dichloroindophenol solution until the pink color was achieved. The titer value was recorded, and then the amount of vitamin C in each sample was calculated.

### 3.6. TOPSIS Analysis

TOPSIS is a classical multicriteria decision-making technique firstly described by Hwang and Lai [64]. Due to its simplicity, rationality, and comprehensibility, this technique has been extensively applied in various researches to rank each option or each alternative based on a set of defined parameters [65,66]. In this study, the TOPSIS method was adopted to assess the effects of MeJA on potato tuber preservation, with hardness, a*, dry matter content, reducing sugar, starch, soluble protein, and vitamin C being very large indexes, and the chlorophyll content, weight loss, sprouting rate, and solanine content being very small indexes.

## 4. Conclusions

When applied at an appropriate dosage on potato tubers, MeJA could significantly suppress sprouting, reduce the accumulation of solanine, alleviate the loss of fresh weight and dry matter, and delay softening and greening. Moreover, MeJA can be successfully encapsulated into an alginate/chitosan polymeric system, thus overcoming the disadvantage of fast deprivation at ambient temperature due to its volatile property and realizing the sustainable release. In comparison with direct spraying, the application of MeJA microcapsules consumes a lower dosage of MeJA but possesses better efficacy in sprouting control and quality maintenance of potato tubers. MeJA microcapsules proved to be promising agents to extend the shelf life of potato tubers with their safety, simplicity, and cost-effectiveness.

## Figures and Tables

**Figure 1 molecules-27-04728-f001:**
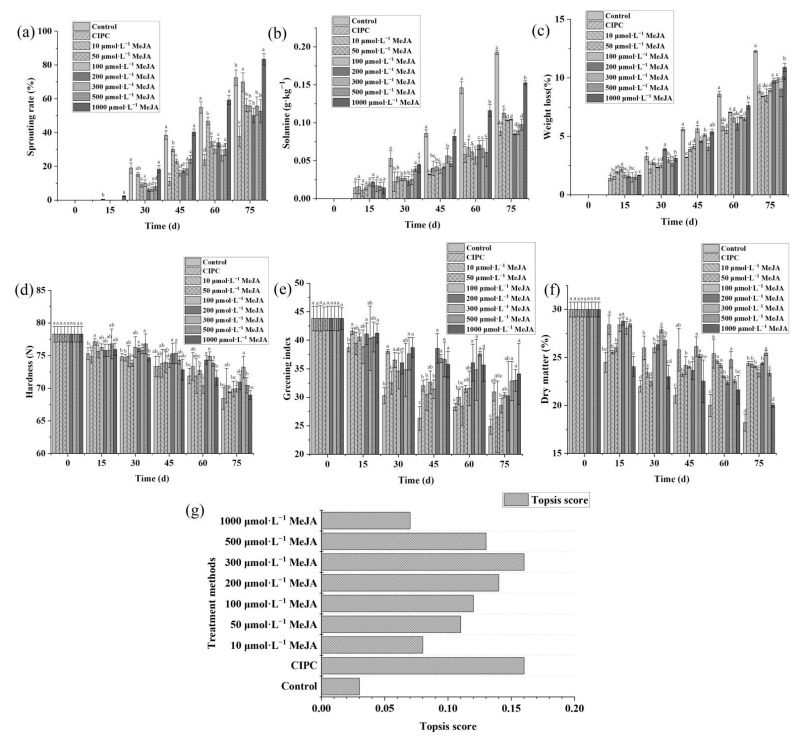
Methyl jasmonate could suppress sprouting and retain the quality of stored potato depending on the applied concentration. Sprouting rate (**a**), solanine content (**b**), weight loss (**c**), hardness (**d**), greening index (**e**), dry matter (**f**), TOPSIS score of the treatments based on these criteria (**g**). Multiple comparisons of the treatments using Turkey’s method (*p* < 0.05) have been conducted, and treatments with no letters in common are significantly different.

**Figure 2 molecules-27-04728-f002:**
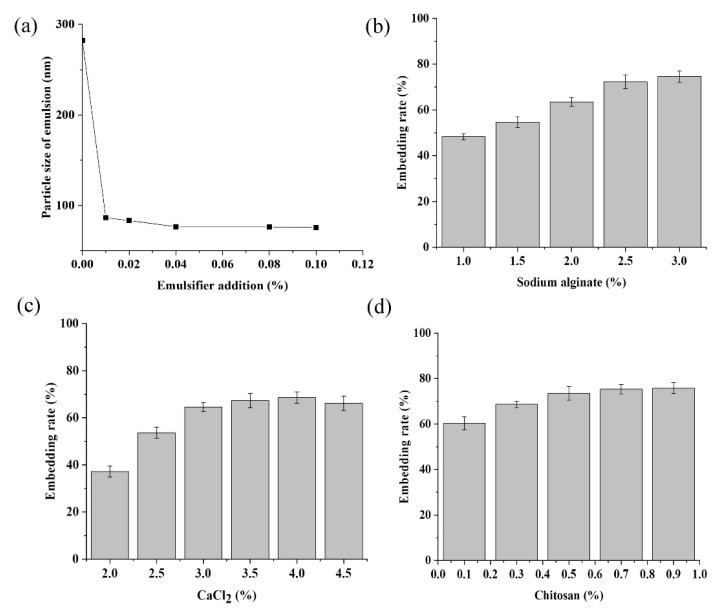
Emulsifier addition could reduce the particle size of methyl jasmonate emulsion (**a**); appropriate concentration of sodium alginate (**b**), CaCl_2_ (**c**), chitosan (**d**) could improve the embedding rate of methyl jasmonate.

**Figure 3 molecules-27-04728-f003:**
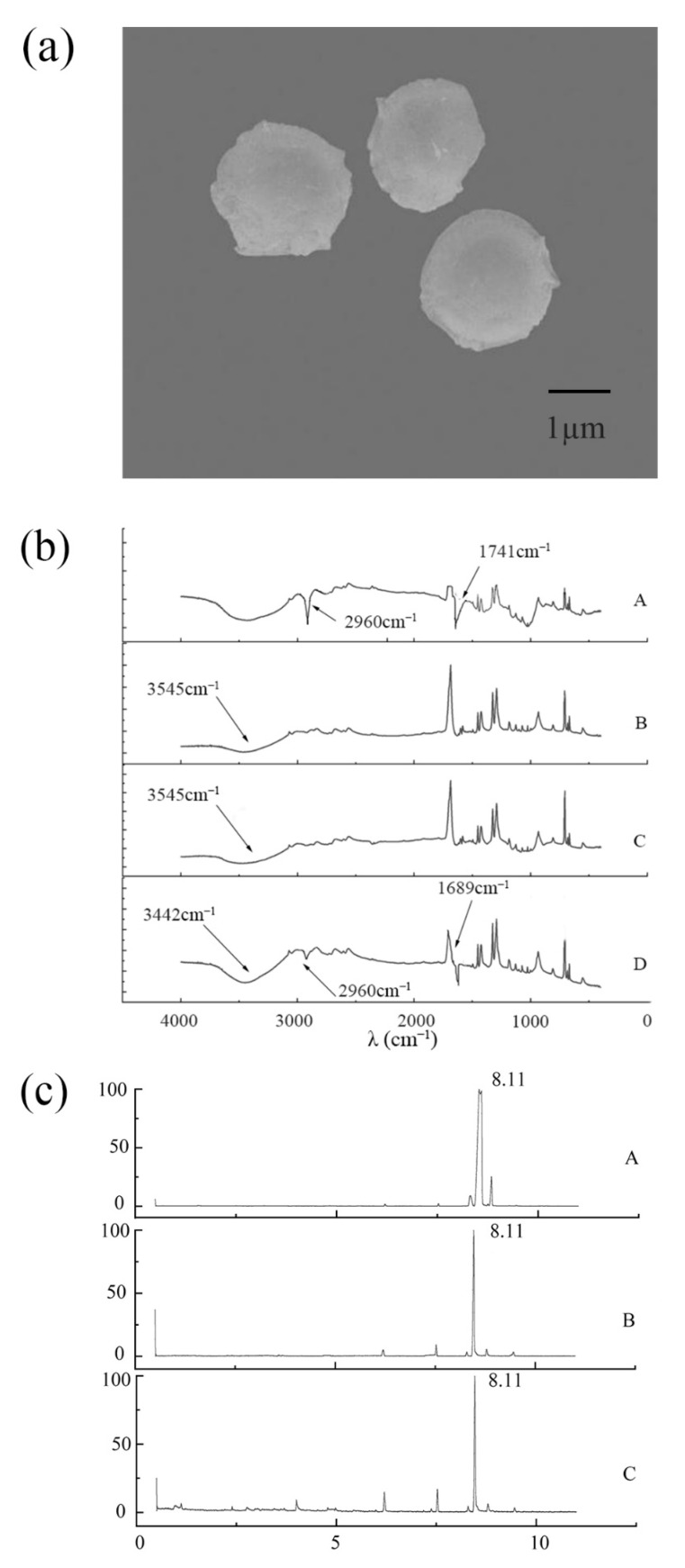
Characterization of methyl jasmonate microcapsules. Morphology of methyl jasmonate microcapsules (**a**); The FT-IR spectrography of pure methyl jasmonate (**b**(A)), mixture of sodium alginate and chitosan (**b**(B)), blank microcapsules (**b**(C)), and MeJA microcapsules (**b**(D)); methyl jasmonate release from methyl jasmonate microcapsules detected by GC-MS at day 0 (**c**(A)), day 15 (**c**(B)) and day 30 (**c**(C)).

**Figure 4 molecules-27-04728-f004:**
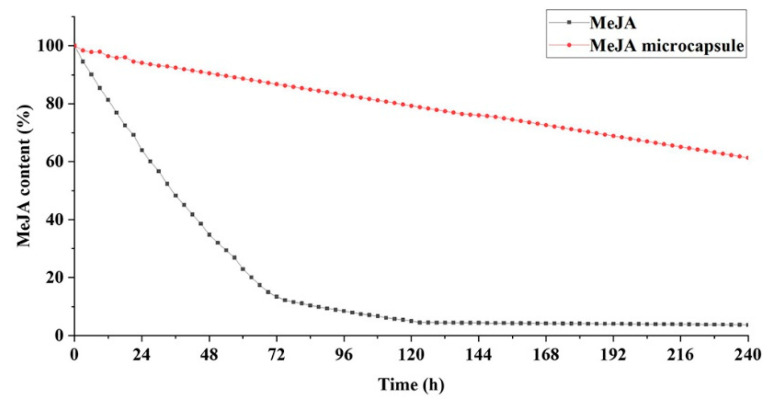
Methyl jasmonate content remained in methyl jasmonate microcapsules over the period of 240 h at 20 °C.

**Figure 5 molecules-27-04728-f005:**
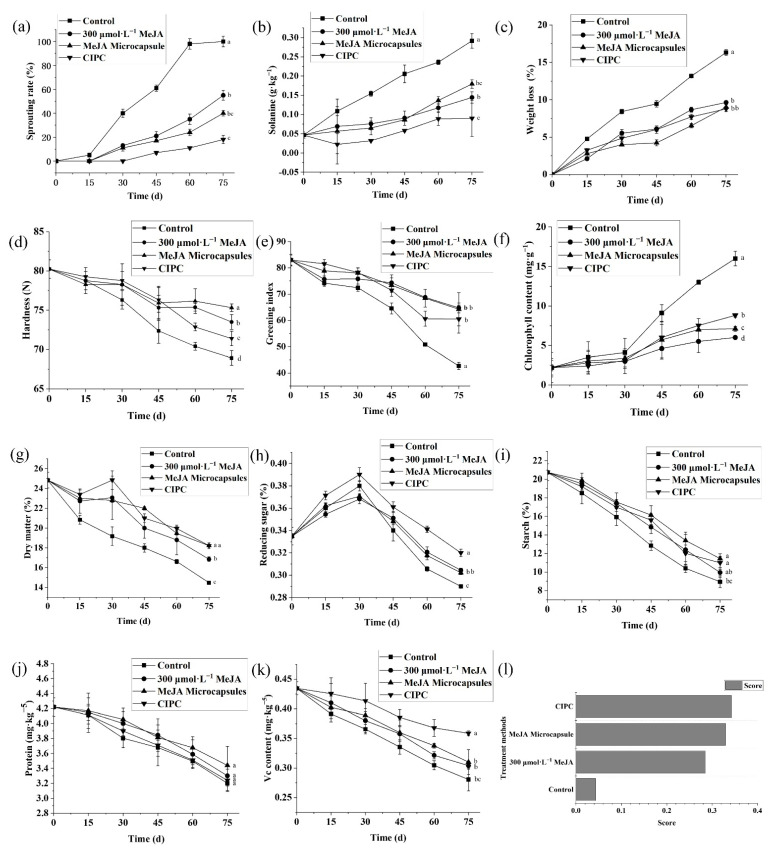
Methyl jasmonate microcapsules have high efficacy on the quality preservation of stored potato. Sprouting rates (**a**), solanine content (**b**), weight loss (**c**), hardness (**d**), greening index (**e**), chlorophyll content (**f**), dry matter (**g**), reducing sugar content (**h**), starch content (**i**), protein content (**j**), vitamin C content (**k**); TOPSIS score of the treatments based on these criteria (**l**). Note: control was treated with blank microcapsules. Multiple comparisons of the treatments using Turkey’s method (*p* < 0.05) have been conducted after 75 d of storage, and treatments with no letters in common are significantly different.

**Table 1 molecules-27-04728-t001:** Orthogonal experimental analysis for the influencing factors in the preparation of methyl jasmonate microcapsules.

Number	Sodium Alginate (%)	CaCl_2_(%)	Chitosan (%)	Embedding Rate (%)
1	1.5	2.5	0.1	67.3
2	1.5	3	0.3	64.5
3	1.5	3.5	0.5	70.3
4	2	2.5	0.3	75.6
5	2	3	0.5	79.3
6	2	3.5	0.1	81.3
7	2.5	2.5	0.5	72.1
8	2.5	3	0.3	83.9
9	2.5	3.5	0.1	80.2
K_1_	202.1	215	228.8	
K_2_	236.2	227.7	224	
K_3_	236.2	231.8	221.7	
k_1_	67.37	71.67	76.27	
k_2_	78.73	75.90	74.67	
k_3_	78.73	77.27	73.90	
R	11.36	5.60	2.37	

K_i_ represents the sum of the embedding rate corresponding to factor level i in the column; k_i_ represents the average of the embedding rate corresponding to factor level i in the column; R represents the difference between the maximum and the minimum of k_i_.

## Data Availability

The data presented in this study are available on request from the corresponding author.

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
