# Peer review of "Preparation and Characterization of Methyl Jasmonate Microcapsules and Their Preserving Effects on Postharvest Potato Tuber"

_molecules, 2022, doi:10.3390/molecules27154728_

Round 1

Reviewer 1 Report

The paper points out differences from related research. The authors targeted the subject of improving the methyl jasmonate permanence and propagation with application on potato tubes. The compound attributes are presented in the literature as an application to other vegetables and/or fruits.

The study has a valuable contribution to considering eco-friendly preserving alternatives. It presents relevant information in the area of ecological approaches.

The Introduction correctly highlights the current concerns in the mentioned domain. Connections to other's work are declared but are supplementations necessary.

The authors are recommended to review the manuscript and pay attention to the following aspects:

1.      Review the English and, where necessary, use the plural form of the substantives.

2.      Go through the Introduction part and include references to sustain important statements (e.g.: ”…. sprouting is considered a significant factor determining its quality and marketability”; ”It has been widely used in food processing, medicine, cosmetics, and many other fields.).

3.      Pay attention to the paragraph's spelling in the Materials part. Between the phrases is ”;” not as should be ”.”.

4.      Different fonts are used in the Results and Discussions part. Please check the aspect.

5.      Include more references in the Results and Discussions to sustain the affirmations made.

  The paper could be accepted for publication after minor changes. The article must be revised by the author(s) and resubmitted with suggested modifications specified in the reviewer’s comments.

Reviewer 2 Report

Changes in the quality characteristics of potatoes during long-term storage are an important element of their commercial quality. The use of microcapsules with a selective and constant MeJA release rate allows for a lower level of adverse changes in potato tubers during storage. MeJA microencapsulation seems to be an alternative to the use of chlorpropham, a preparation with a proven negative effect on human health. In the presented manuscript, the authors described the possibilities of using microencapsulated MeJA with selective release on the quality parameters of potato tubers during 75-day storage. However, some corrections are required in the manuscript. Remarks

2.1 At what stage of maturity were they harvested

Convert the concentration of the solution to the aqueous concentration of the solution

2.3.1 The morphology of the capsules was carried out at what magnification

2.3.2 Microcapsules were mixed with KBr, define the abbreviation

2.3.3 describe the method of application to the column

2.4 indicate the storage temperature of the potatoes

2.5.2 indicate which detector has been worked on

2.5.3 indicate when the greening was measured

2.5.4 report when the chlorophyll has been measured

2.5.5 which device was used to measure the hardness

3 what amounts of MeJA in each concentration were consumed

No statistical analysis of the obtained results,

The concentration of MeJA in capsules is given, but the amounts used in the studies are not given

There is no description of changes in vitamin C levels in the discussion

Give the models of the chromatographs used
